# Discovery of an exosite on the SOCS2-SH2 domain that enhances SH2 binding to phosphorylated ligands

Edmond M. Linossi[1,2,6], Kunlun Li[1,2,6], Gianluca Veggiani[3,6], Cyrus Tan[1,2], Farhad Dehkhoda[1,2], Colin Hockings[1,2], Dale J. Calleja [1,2], Narelle Keating[1,2], Rebecca Feltham [1,2], Andrew J. Brooks [4], Shawn S. Li [5], Sachdev S. Sidhu [3], Jeffrey J. Babon [1,2], Nadia J. Kershaw [1,2,7✉] & Sandra E. Nicholson [1,2,7✉]

Suppressor of cytokine signaling (SOCS)2 protein is a key negative regulator of the growth hormone (GH) and Janus kinase (JAK)-Signal Transducers and Activators of Transcription (STAT) signaling cascade. The central SOCS2-Src homology 2 (SH2) domain is characteristic of the SOCS family proteins and is an important module that facilitates recognition of targets bearing phosphorylated tyrosine (pTyr) residues. Here we identify an exosite on the SOCS2-SH2 domain which, when bound to a non-phosphorylated peptide (F3), enhances SH2 affinity for canonical phosphorylated ligands. Solution of the SOCS2/F3 crystal structure reveals F3 as an α-helix which binds on the opposite side of the SH2 domain to the phosphopeptide binding site. F3:exosite binding appears to stabilise the SOCS2-SH2 domain, resulting in slower dissociation of phosphorylated ligands and consequently, enhances binding affinity. This biophysical enhancement of SH2:pTyr binding affinity translates to increase SOCS2 inhibition of GH signaling.

[1] The Walter and Eliza Hall Institute of Medical Research, Parkville, VIC, Australia. [2] Department of Medical Biology, University of Melbourne, Parkville, VIC, Australia. [3] The Donnelly Center for Cellular and Biomolecular Research, University of Toronto, Toronto, ON, Canada. [4] The University of Queensland Diamantina Institute, Woolloongabba, QLD 4102, Australia. [5] Department of Biochemistry and the Siebens-Drake Medical Research Institute, Schulich School of Medicine and Dentistry, University of Western Ontario, London, Canada. [6] These authors contributed equally: Edmond M. Linossi, Kunlun Li, Gianluca Veggiani. [7] These authors jointly supervised this work: Nadia J. Kershaw, Sandra E. Nicholson. ✉email: kershaw@wehi.edu.au; snicholson@wehi.edu.au

Tyrosine phosphorylation is a fundamental step in many signal transduction cascades required to mediate cellular processes. The Src homology 2 (SH2) domain forms the largest family of phosphotyrosine recognition domains, with a total of 120 different SH2 domains found in 110 distinct proteins in the human genome[1]. The binding specificities between different SH2 domains and the sequence surrounding the phosphotyrosine motif in individual proteins guarantee the accurate recognition of the target protein and tightly regulated signal transmission. The SH2 domain is a small protein module consisting of three β-strands flanked by two α-helixes. These secondary structural elements and the loops that connect them form two pockets; one is the highly conserved phosphotyrosine binding pocket, and the second is a hydrophobic pocket that typically accommodates hydrophobic residues in the target protein located in the "+3" position following the phosphotyrosine[2,3].

Members of the Suppressors of Cytokine Signaling (SOCS) protein family are the primary negative regulators of Janus kinase (JAK)-signal transducer and activator of transcription protein (STAT) signaling. Eight members have been identified, including CIS and SOCS1-7, which are defined by a central SH2 domain and a C-terminal SOCS box motif[4,5]. The SOCS-SH2 domain conforms to the canonical SH2 domain structural fold and is distinguished by an additional α-helix N-terminal to the domain, designated the extended SH2 subdomain (ESS)[6–8]. Although some phosphotyrosine motif binding preferences have been defined for individual SOCS-SH2 domains[9–12], the full suite of protein targets and preferred binding sequences for each SOCS-SH2 domain remain to be elucidated. The SOCS proteins themselves have clearly defined biological roles in regulating signaling cascades activated by different cytokines and growth factors[13–19]. The SOCS-SH2 domain is central to their function and in all instances tested, loss of SH2 binding to phosphotyrosine causes loss of SOCS activity[20–22]. The SOCS box, a conserved 40-residue motif at the C-terminus of the SH2 domain, forms two alpha-helices and is constitutively associated with the adaptor proteins Elongin B and Elongin C, which function to both stabilize the SH2 domain and recruit an E3 ubiquitin ligase complex consisting of Cullin5 and RING-box protein (Rbx)2[5,23]. This recruitment allows the ubiquitination and proteasomal degradation of targets bound to the SOCS-SH2 domains and classes the SOCS proteins as Cullin-RING E3 ligases (CRLs)[23].

SOCS2 is the predominant negative regulator of growth hormone (GH) and prolactin signaling[13,19,24–26]. Consequently, SOCS2-deficient mice (Socs2−/−) display gigantism, weighing significantly more than their wild-type counterparts at 6 weeks of age, due to enhanced GH signaling[19]. SOCS2 regulation of GH signaling is thought to primarily occur via SOCS2-SH2 domain recognition of phosphotyrosine Tyr595 in the GH receptor (GHR)[13]. However, the paradoxical observation that mice expressing a Socs2 transgene also display gigantism, remains mechanistically unexplained[24]. Several reports indicate an important role for SOCS2 in immune cells, with Socs2−/− mice displaying hyper-responsiveness to microbial stimuli in dendritic cells (DCs) and skewed T helper type 2 (Th2) differentiation and impaired generation of inducible regulatory T cells (iTregs)[27–29]. In addition, Socs2 is upregulated by interferon-gamma (IFNγ) in DCs in human melanoma, and Socs2−/− mice exhibit enhanced DC priming of Tcell responses to give improved antitumor immunity[30].

SOCS2 has been reported to inhibit both STAT activation[31] and NF-κB signaling; regulating the nuclear translocation of the NF-κB p65 subunit in DCs[32], with primary Socs2−/− hepatocytes showing increased phospho-p65 in the nucleus compared with wild-type cells[33]. SOCS2 clearly has many diverse roles in biological processes that are important for cancer, autoimmunity, and other inflammatory diseases. Understanding the function and recognition specificity of the SOCS2-SH2 domain is critical to understanding how SOCS2 regulates these different signaling cascades.

Here, we have discovered an exosite on the SOCS2-SH2 domain that, when occupied by a peptide (F3), reciprocally enhances the affinity of the SH2 domain for its canonical phosphorylated ligands. X-ray crystallography and NMR analysis revealed F3-bound as an α-helix on the opposite side to the phosphotyrosine binding pocket. F3 binding appeared to stabilize the ESS helix and phosphotyrosine binding loop, resulting in a reduced dissociation rate and increased SH2 affinity for its phosphorylated protein targets. Apart from the closely related CIS-SH2 domain, F3 binding was highly selective for SOCS2, with CIS displaying similar, although reduced, enhancement of affinity for phosphorylated ligands. This suggests the F3:exosite phenomenon might be applicable to other SH2 domains, with the cooperative increase in binding affinity implying an unappreciated degree of regulation. Expression of F3 peptide in A549 cells enhanced SOCS2-SH2 binding to the GHR and increased SOCS2 inhibition of GH signaling. Consequently, biophysical measurement of SH2-binding affinities might be substantially underestimated, particularly if both exosite and canonical sites are simultaneously occupied by proteins in the same complex.

## Results

**Identification of non-phosphorylated peptides which bind to the SOCS2-SH2 domain.** In order to identify peptides that could bind to the surface of the SOCS2-SH2 domain, we used a highly degenerate phage-displayed library encoding random hexadecapeptides containing combinations of all 19 genetically encoded amino acids except cysteine[34]. The peptide library was cycled through five rounds of selection for binding to a fusion protein consisting of the SOCS2-SH2 domain and SOCS box fused to the C-terminus of glutathione S-transferase (GST), in complex with elongins B and C (GST-SOCS2$^{32–198}$-EloB/C; SOCS2 lacking the N-terminal region) (Fig. 1a). To maximize the chances of isolating highly specific peptides, prior to binding to the GST-SOCS2$^{32–198}$-EloB/C complex, the library was incubated with bovine serum albumin (BSA) and GST to deplete the phage pool of non-specific peptides. Following biopanning, we used phage-ELISAs to detect clones that showed binding to the GST-SOCS2$^{32–198}$-EloB/C complex, but not to BSA or GST, and subjected specific clones to DNA sequence analysis. Selections yielded three unique non-phosphorylated peptides that bound with high affinity to the GST-SOCS2$^{32–198}$-EloB/C complex and contained a homologous central sequence motif (HLMxKW). Synthetic peptides bound to SOCS2$^{32–198}$-EloB/C with $K_D$ 1.68–19.33 μM, as measured by isothermal calorimetry (ITC) (Fig. 1b; Supplementary Table 1).

A series of alanine point mutations of the highest affinity peptide (F3; 1.68 μM) confirmed that the central sequence motif was absolutely required for binding, whereas mutation to residues with similar physio-chemical properties, indicated some tolerance of related residues (with the exception of K10H) (Fig. 1b and Table 1). The residues N- and C-terminal to the HLMxKW motif also appeared to contribute to binding, given that the three homologous peptides displayed varying affinities, and that truncation of more than two residues from each terminus of the F3 peptide was sufficient to abrogate binding (Table 1).

F3 did not interact with the SOCS2-SOCS box complex alone (SOCS2$^{157-198}$-EloB/C; Supplementary Table 1), nor did the presence of Cullin5 in the complex impact on F3 binding (Supplementary Fig. 1), indicating that F3 interacted directly with the SOCS2-SH2 domain.

**The F3 peptide allosterically enhances SOCS2-SH2 domain binding to phosphorylated proteins.** We initially hypothesized that the F3 peptide would bind the canonical phosphopeptide binding interface; however, the F3 peptide did not compete with a

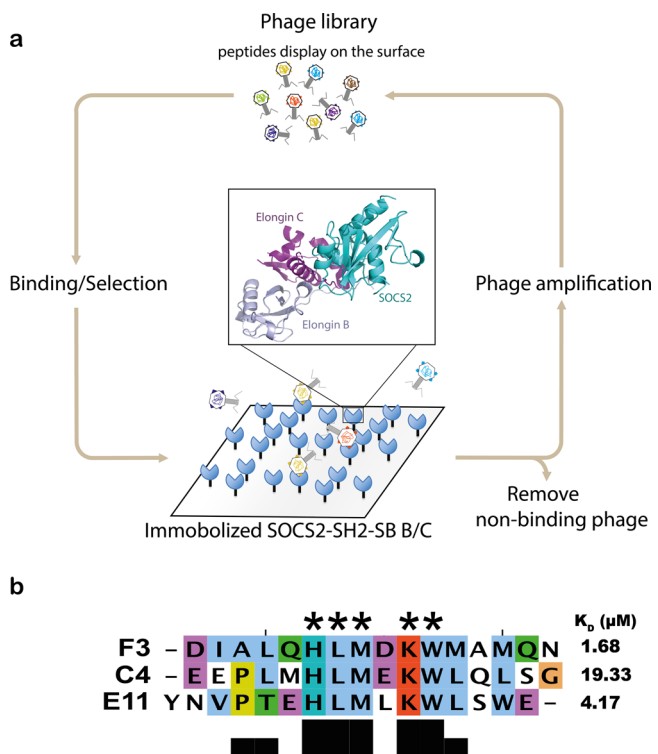

**Fig. 1 Identification of a peptide that binds with low micromolar affinity to the SOCS2/Elongin B/C complex. a** Phage display of a degenerate peptide library was used to enrich for non-phosphorylated peptides that bound to SOCS2. GST-SOCS2-SH2-SOCS box protein was produced as a trimeric complex in *E. coli* by co-expression with elongins B and C. Protein was produced with an N-terminal GST tag and lacks the SOCS2-N-terminal region [GST-SOCS2$^{32-198}$-EloB/C]. **b** Three peptide sequences that shared amino-acid sequence homology were enriched by phage display. Colors indicate amino-acid similarity. *indicates mutation to Ala = loss of binding. ITC was used to determine the binding affinities ($K_D$) of the individual peptides (shown on the right) (related to Table 1).

tyrosine-phosphorylated peptide (pY595) derived from the GHR. Instead, the presence of F3 unexpectedly enhanced SOCS2 binding to the GHR pY595 peptide (~20-fold increase from 0.70 μM to 0.04 μM) (Fig. 2a). The effect was reciprocal, with the GHR peptide, in turn, enhancing SOCS2 binding to F3, in addition to the homologous E4 and F11 peptides (Fig. 2b, d and Supplementary Table 1). The impact of F3 on SOCS2-SH2 affinity for GHR pY595 was further validated by competition SPR (~10-fold increase; Supplementary Fig. 2a). The difference between ITC and SPR (0.04 c.f. 0.15 μM in the presence of F3) was within acceptable experimental variation, and subsequent experiments were performed using SPR owing to its higher throughput and sensitivity. The F3-driven enhancement was not restricted to the GHR pY595 peptide, as SOCS2 also displayed an enhanced affinity for tyrosine-phosphorylated peptides derived from murine gp130 pY757 and erythropoietin (EPO) receptor Y426 (Supplementary Table 1). We also examined the F3:SOCS2 interaction using a thermal stability assay. F3 binding showed a clear stabilization effect on both the apo and phosphopeptide-bound forms, as expected. F3 binding increased the melting temperature (TM) of SOCS2 by 5.8 degrees, whereas binding of a phosphorylated ligand increased the TM by 13.0 degrees. The combination of F3 and phosphopeptide resulted in a further increase in TM, which was not completely additive (Supplementary Fig. 2b). Given that the SOCS2 construct used in SPR and ITC experiments lacked the N-terminal region, we verified that F3 bound to full-length SOCS2$^{1-198}$-EloB/C with a similar affinity and enhancement of phosphopeptide binding, as that observed for the truncated SOCS2$^{32-198}$-EloB/C construct (Supplementary Fig. 2a). This suggests that the N-terminal region does not occupy the allosteric site on the SOCS2-SH2 domain.

To confirm that F3 enhanced binding of the SOCS2-SH2 domain to intact phosphorylated proteins, HepG2 cells were treated with pervanadate to increase global phosphorylation, lysed and GST or GST-SOCS2$^{32-198}$-EloB/C used to enrich phosphorylated proteins. GST-SOCS2$^{32-198}$-EloB/C specifically enriched tyrosine-phosphorylated proteins (identified by GHR pY595 competition) and this was enhanced in the presence of the F3 peptide, including detection of additional interacting phospho-proteins (Fig. 2c). We next investigated F3:SOCS2 interaction in live HEK293T cells, using fluorescence lifetime imaging microscopy (FLIM). Representative images showed lower fluorescence lifetimes in cells expressing SOCS2 and mRuby3-F3, compared with cells expressing only SOCS2, or SOCS2 and mRuby3

| Table 1 Mutation of F3 peptide impacts on binding to SOCS2. | | | | | | | | | | | | | | | | |
|---|---|---|---|---|---|---|---|---|---|---|---|---|---|---|---|---|
| **Peptides** | **Sequences** | | | | | | | | | | | | | | | **KD (μM) IC$_{50}$ (μM)\*** |
| F3 | D | I | A | L | Q | H | L | M | D | K | W | M | A | M | Q | N | 4.19 ± 0.22 |
| H6A | D | I | A | L | Q | **A** | L | M | D | K | W | M | A | M | Q | N | >10 |
| L7A | D | I | A | L | Q | H | **A** | M | D | K | W | M | A | M | Q | N | NB |
| M8A | D | I | A | L | Q | H | L | **A** | D | K | W | M | A | M | Q | N | NB |
| D9A | D | I | A | L | Q | H | L | M | **A** | K | W | M | A | M | Q | N | 3.18 ± 0.43 |
| K10A | D | I | A | L | Q | H | L | M | D | **A** | W | M | A | M | Q | N | NB |
| W11A | D | I | A | L | Q | H | L | M | D | K | **A** | M | A | M | Q | N | >10 |
| F3 | D | I | A | L | Q | H | L | M | D | K | W | M | A | M | Q | N | 0.67 ± 0.03\* |
| H6K | D | I | A | L | Q | **K** | L | M | D | K | W | M | A | M | Q | N | 1.20 ± 0.01\* |
| L7V | D | I | A | L | Q | H | **V** | M | D | K | W | M | A | M | Q | N | 2.02 ± 0.48\* |
| M8L | D | I | A | L | Q | H | L | **L** | D | K | W | M | A | M | Q | N | 0.77 ± 0.32\* |
| K10H | D | I | A | L | Q | H | L | M | D | **H** | W | M | A | M | Q | N | >10\* |
| W11F | D | I | A | L | Q | H | L | M | D | K | **F** | M | A | M | Q | N | 2.23 ± 1.26\* |
| ΔN2ΔC2 | – | – | A | L | Q | H | L | M | D | K | W | M | A | M | – | – | 2.84 ± 0.38\* |
| ΔN5 | – | – | – | – | – | L | M | D | K | W | M | A | M | Q | N |  | NB |
| ΔC4 | D | I | A | L | Q | H | L | M | D | K | W | M | – | – | – | – | NB |
| ΔN3ΔC3 | – | – | – | L | Q | H | L | M | D | K | W | M | A | – | – | – | NB |

F3 peptide binding to SOCS2 was measured by ITC and SPR; ITC values are average ± range, derived from two independent experiments. *SPR values are mean ± S.D., derived from three independent experiments. Bold highlights residue changes. *NB* no binding. SPR curves are shown in Supplementary Fig. 10a. Source data are provided as a Source Data file.

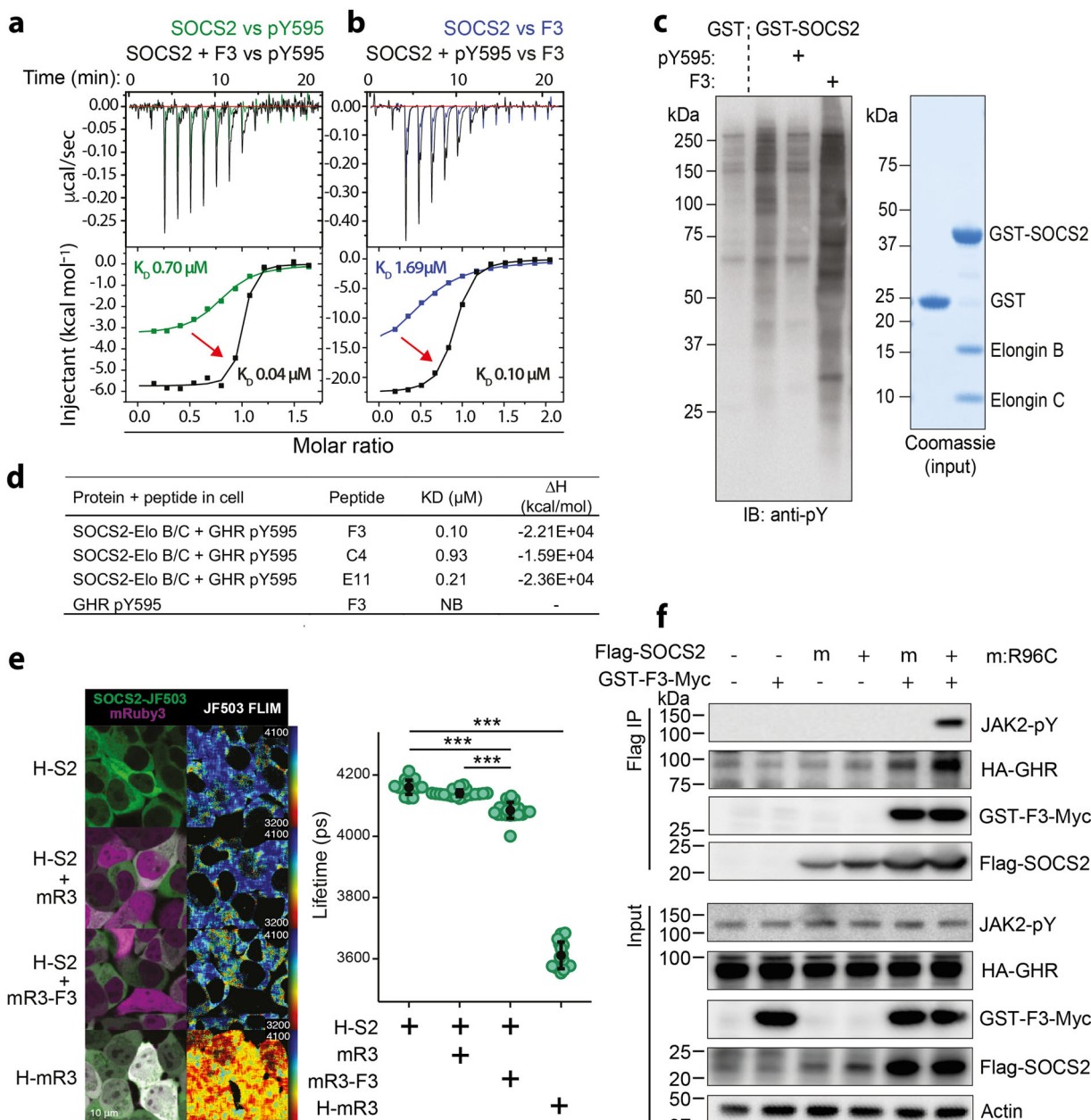

**Fig. 2 The SOCS2-SH2 domain contains an exosite that enhances its ability to bind phosphorylated ligands.** Representative ITC curves showing **a** enhanced binding of the SOCS2[32–198]-EloB/C complex to a phosphorylated peptide derived from the GHR (pY595) in the presence of the F3 peptide (black: + F3; green: no F3) and **b** enhanced binding of the SOCS2 B/C complex to F3 in the presence of the GHR pY595 peptide (black: +pY595; blue: no pY595). **c** SOCS2 affinity enrichment. HepG2 cells were treated with pervanadate to increase phosphorylation, lysed, and incubated with recombinant GST or GST-SOCS2[32–198]-EloB/C (GST-SOCS2) in the presence or absence of the GHR pY595 and F3 peptides. GST-enriched protein complexes were assessed by immunoblot (IB) with anti-phosphotyrosine antibodies (anti-pY; left panel). Protein input by Coomassie gel (right panel). Representative of two independent experiments. **d** Binding affinities between the three homologous non-phosphorylated peptides identified by phage display, and SOCS2[32–198]-EloB/C pre-incubated with GHR pY595, assessed by ITC. Related to Supplementary Table 2. *NB* no binding detected **e** Halo-SOCS2 (H-S2) labeled with JF503 showed specific FLIM-FRET interaction with mRuby3-F3-Myc fusion protein (mR3-F3). A FRET interaction shortens the fluorescence lifetime of the donor (JF503). HEK293T cells expressing Halo-SOCS2 and/or mRuby3-F3-Myc (or mRuby3-Myc), or positive control Halo-mRuby3 (H-mR3), as indicated, were stained with Halo ligand JF503 and imaged by fluorescence lifetime imaging microscopy (FLIM). Lefthand side images show fluorescence intensity. Righthand side images (FLIM) show lower fluorescence lifetimes (rainbow scheme skewed towards red/yellow) in cells expressing SOCS2 and mRuby3-F3-Myc, than cells expressing only SOCS2, or SOCS2 and mRuby3 without F3. Images displayed are identical with respect to instrumentation, analysis and display settings, and are representative of three independent experiments. FLIM images are displayed with a rainbow color scale, with lower and upper limits of 3200 ps and 4100 ps (lifetime), respectively. The plot shows lifetime fluorescence as mean ± S.D. $n = 18$–23 images collated from three independent experiments (5–12 images from each experiment). *$p < 0.001$ (one-way ANOVA with Tukey HSD post hoc test). $H$ = Halotag. Additional images are shown in Supplementary Fig. 3. **f** F3 enhances SOCS2 binding to GHR. 293T-GHR cells were transfected with constructs expressing Flag-SOCS2 and GST-F3-Myc and treated with 50 ng/mL growth hormone (GH) prior to cell lysis. Anti-Flag-SOCS2 immunoprecipitates were analyzed by immunoblotting for interaction with GST-F3-Myc, GHR, and tyrosine-phosphorylated (pY) proteins using the appropriate antibodies. Representative of two independent experiments. Related to Supplementary Figs. 4 and 7. Source data are provided as a Source Data file.

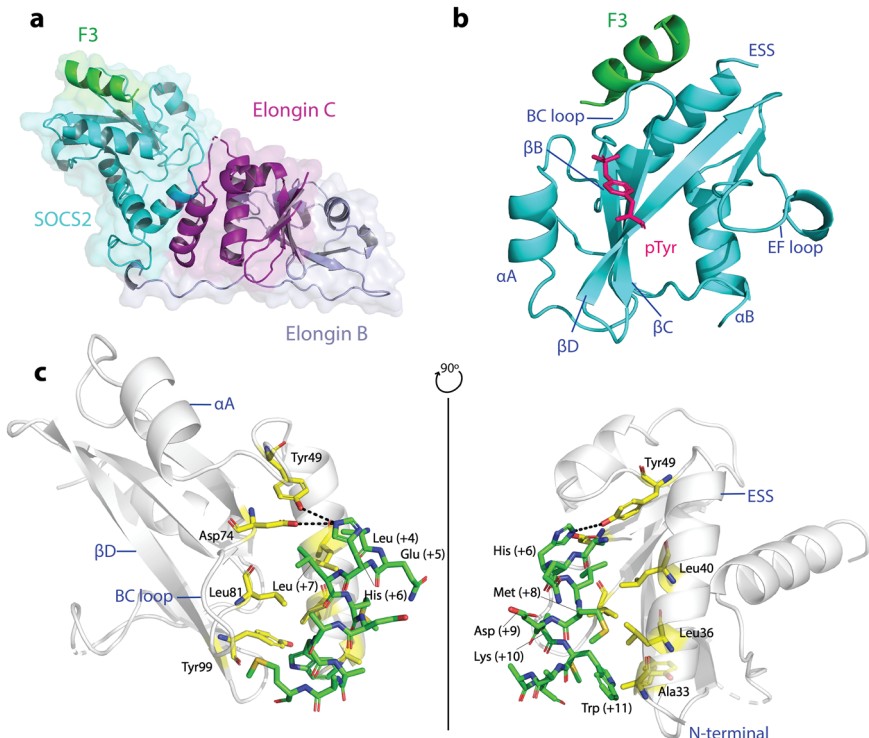

**Fig. 3 The crystal structure of F3 in complex with SOCS2, reveals F3 as an α-helix bound to a site distal from the pTyr-binding pocket. a** Cartoon and surface representation of the F3 (green)/SOCS2 (cyan)/Elongin B (light blue-purple)/Elongin C (magenta) 3.1 Å crystal structure. **b** Notably, F3 binds as an α-helix on the opposite side to the pTyr-binding pocket, cradled between the ESS and phosphotyrosine binding loop. **c** Ribbon diagram of the SOCS2-SH2 (white) and F3 structure showing the detailed side-chain interactions. The SOCS2-ESS and canonical SH2 domain make multiple contacts with the F3 peptide. Seven key residues within the SOCS2-ESS, the loop connecting ESS and α-helix A, the BC loop, and β-strand D, make key contacts with F3 residues and are highlighted in yellow. Dashed lines indicated polar interactions between F3 (His6) and residues on SOCS2 F3 peptide: carbon = green; oxygen = red; nitrogen = blue. Figures were generated using The PyMOL Molecular Graphics System, Version 2.0 Schrödinger, LLC.

(without F3), indicating that SOCS2 interacts with F3 in intact cells (Fig. 2e and Supplementary Fig. 3). To further confirm that binding to phosphorylated proteins was enhanced by F3, 293T-GHR cells were co-transfected with constructs encoding Flag-SOCS2 and GST-F3-Myc and treated with GH prior to lysis. Immunoblot analysis of anti-Flag immunoprecipitates indicated that GST-F3-Myc is efficiently bound to Flag-SOCS2, enhancing its ability to enrich GHR and phosphorylated JAK2 (Fig. 2f). Mutation of R96C in the SOCS2-SH2 domain, disrupts binding to phosphopeptide[20], but not F3 (Supplementary Fig. 7), and was used to confirm that SOCS2 enrichment of GHR and JAK2 was phosphotyrosine dependent. Conversely, GST-F3 affinity enrichment further validated the interaction with SOCS2 and showed an enhanced association of multiple tyrosine-phosphorylated proteins in the presence of SOCS2 (Supplementary Fig. 4).

**F3 binds as an alpha-helix in a unique exosite on the SOCS2-SH2 domain**. To understand how F3 binding to the SOCS2-SH2 domain enhanced its interaction with phosphorylated target proteins, structural studies were performed using both NMR and X-ray crystallography. A SOCS2 construct with a cluster of three mutations K115A/K117A/Q118A (SOCS2$^{32–198}$-KKQ) previously reported to reduce surface conformational entropy and improve crystallization[35] was used, in complex with elongins B and C, and bound to the shortest form of F3 that retained binding (residues 3–14, ALQHLMDKWMAM; Table 1). Crystals of a 1:1:1:1 complex (F3:SOCS2:EloB:EloC) were obtained and data processed to 3.1 Å. The structure was determined by molecular replacement using the SOCS2/Elongin BC as a model (PDB:

2C9W[6]) and refined to $R_{work}$ and $R_{free}$ values of 0.2271 and 0.2667, respectively (Supplementary Table 5). The overall structure of the SOCS2-SH2 domain and SOCS box in complex with elongins B and C (Fig. 3a) accorded well with previously published structures of SOCS2[35], with an R.M.S.D of 0.6 Å across all Cα (DALI) apo SOCS2 (PDB: 5BO4 chain A) and 1.1 Å when aligned with a phosphopeptide-bound structure (PDB: 6I4X) (Supplementary Fig. 6).

F3 bound as an α-helix on the opposite side to the phosphopeptide binding site, in a shallow groove formed by the pTyr-binding loop (defined as the BC loop) and the ESS helix (Fig. 3). Residues 4–14 of the peptide were visible and the buried surface area of F3 when in contact with SOCS2 is 400Å² (PISA[36]). The interaction surface is largely hydrophobic. The side chains of Ala33, Leu36, Ala37, Leu40 from the ESS helix and Leu81, Leu82, and Tyr99 from the SH2 domain, form a cradle to complement the hydrophobic face of the F3 helix, consisting of F3 residues Leu7, Met8, and Trp11. The interaction of the Met14 side chain on F3 with the peptide bond between SOCS2-D79 and Tyr80 further extends the surface. In addition, F3-His6 forms a polar network with side chains of Tyr49 and Asp74, although the resolution precludes a precise mapping of the interactions (Fig. 3c).

The crystallographic data was strongly supported by NMR analysis, with Ala34, Arg41, Glu42 (ESS), Ser75, Ser78, Tyr80 (BC loop), and Gly102 and Val122 in SOCS2 showing significant chemical shift perturbations (CSPs) upon F3 binding[37] (Fig. 4a, b).

The structure was further validated by mutagenesis of both the peptide (Table 1) and SOCS2 (Table 2). An alanine scan of F3 peptide residues from positions 6 to 11 showed that all residues had a significant effect on binding affinity, with the exception of

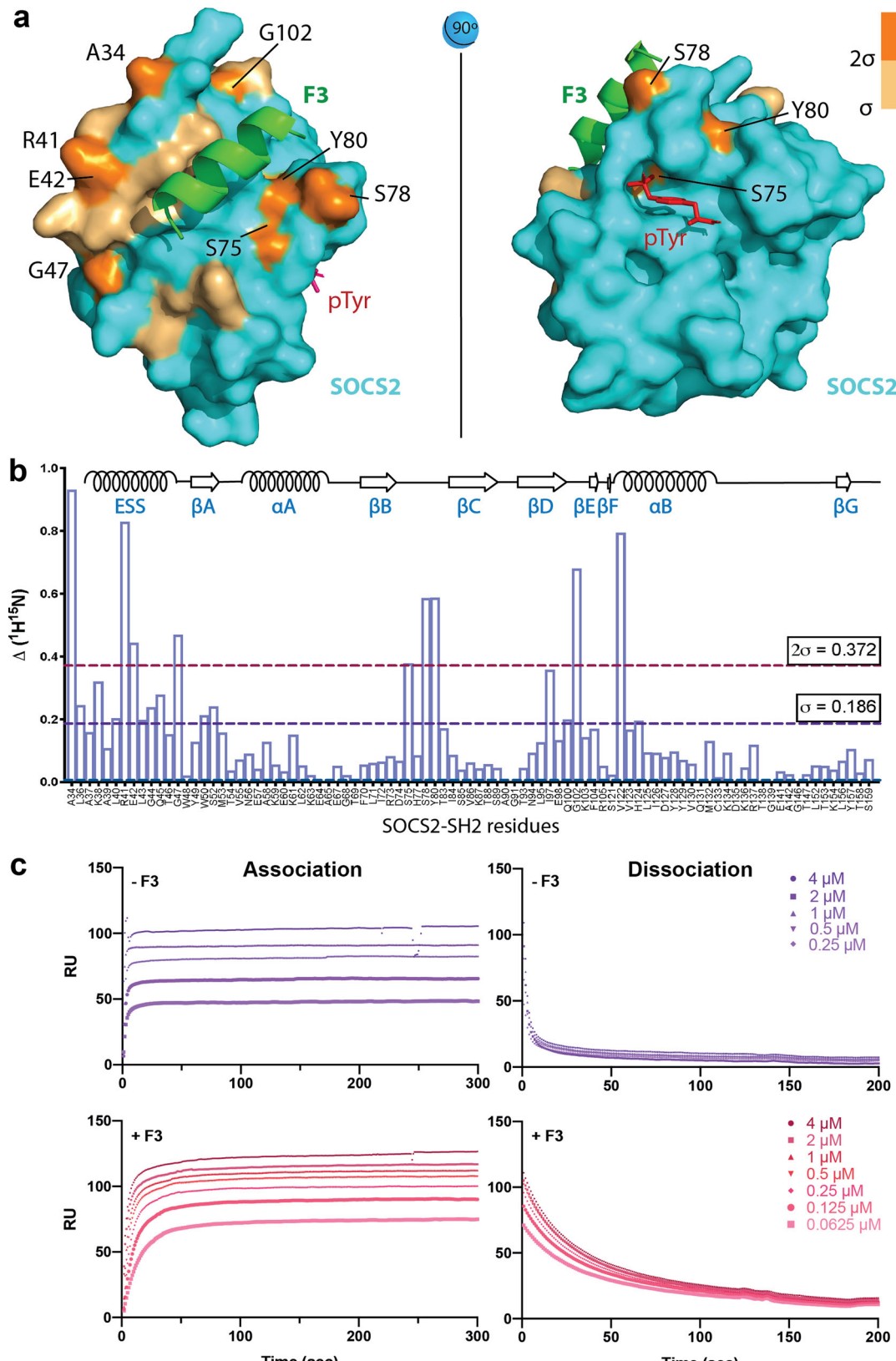

Asp9, which is the only residue in this region that does not contact SOCS2, but rather points into solution. Mutation of the hydrophobic residues Leu7, Met8, and Trp11 to smaller hydrophobic residues retained binding, indicating that the hydrophobic surface is not significantly altered by these more conservative mutations.

Two key polar residues were identified in F3, with alanine substitutions of His6 and Lys10 abrogating binding. His6 cannot be substituted for alanine, but can be substituted for lysine, which can presumably replicate the polar interactions with SOCS2-Tyr49 and Asp74. Although electron density for the full Lys10 side chain is not visible, the most favorable

**Fig. 4 The crystallographic F3/SOCS2³²⁻¹⁹⁸-EloB/C interface is supported by NMR analysis. a** F3-bound SOCS2 crystal structure (cyan) with residues displaying NMR chemical shift perturbations on F3 binding, mapped on the surface. Chemical shift perturbations caused by F3 binding are highlighted in light orange (>σ) and orange (>2σ), corresponding to Fig. 4b. Orange residues (>2σ) are labeled in black. pTyr location corresponds to PDB:6I4X. **b** $^1H^{15}N$ chemical shift perturbation of individual SOCS2 residues on F3 binding. Chemical shift perturbations (CSPs) are calculated as $\Delta^1H$ and $\Delta^{15}N$ of assigned backbone residues of SOCS2-SH2³²⁻¹⁵⁹. σ represents the standard deviation of all calculated chemical shift perturbations. CSPs above σ are considered significant. Sigma: 1 standard deviation of all chemical shift changes. Sigma and two-sigma were used as cutoffs to identify residues with the greatest CSP. **c** F3 binding reduces the dissociation rate of pTyr peptide. On and off rates in the presence and absence of F3. SPR with a SOCS2 titration was performed to observe GHR pY595 peptide association (left panels) and dissociation (right panels) curves. The $k_{off}$ (dissociation rate) is 0.245 ± 0.003 sec⁻¹ (-F3) and 0.0235 ± 0.001 sec⁻¹ (+F3). *RU* response units. Data shown are from a representative experiment. All replicates and data fitting are shown in Supplementary Fig. 10b. Source data are provided as a Source Data file.

**Table 2 Mutation of SOCS2 impacts the binding of F3, but not GHR pY595 peptide.**

| | Location | GHR pY595 (μM) | F3 (μM) | GHR pY595 + F3 (μM)* |
|---|---|---|---|---|
| **SOCS2 mutation** | | | | |
| WT | ESS | 1.08 ± 0.07 | 0.67 ± 0.03 | 0.15 ± 0.02 |
| L36A | | 0.80 ± 0.02 | ND[1] | 0.50 ± 0.02 |
| L40F | | 1.03 ± 0.07 | 0.81 ± 0.02 | 0.29 ± 0.02 |
| L40A | | 1.76 ± 0.21 | >50 | 0.46 ± 0.08 |
| R41A | | 1.18 ± 0.54 | 0.89 ± 0.05 | 0.25 ± 0.07 |
| Q45A | | 1.32 ± 0.03 | 0.60 ± 0.34 | 0.22 ± 0.03 |
| Y49F | βA | 0.98 ± 0.01 | 0.79 ± 0.01 | 0.18 ± 0.01 |
| Y99A | βD | 2.03 ± 0.05 | ND | 1.84 ± 0.4 |
| L81F | βC | 1.88 ± 0.01 | ND | 1.75 ± 0.27 |
| L82F | βC | 1.17 ± 0.04 | 1.53 ± 0.03 | 0.45 ± 0.02 |
| L81, 82F | βC | 0.93 ± 0.34 | ND | 1.94 ± 0.27 |

IC₅₀ values were determined by SPR; [1]ND=not detectable.
*Enhanced GHR pY595 affinities were measured using 10 μM F3 and 100 nM SOCS2. Values are mean ± S.D., derived from 3 independent experiments.

rotamer for lysine would position the terminal amine at an ideal distance to interact with the cluster of backbone carbonyls formed from residues His77 and Tyr80 and both side chain and carbonyl of Ser75. This is consistent with the loss of binding upon mutagenesis of Lys10 to alanine, but slight retention of binding when substituted for histidine, which while shorter, might still be able to hydrogen bond to nearby carbonyls such as that of Ser78. NMR data showing chemical shift perturbations in this region upon F3 binding is also supportive of this interaction, with Arg41, Ser75, Ser78, Tyr80, Gly102, and Val122 in SOCS2 showing significant CSPs (>2σ) on F3 binding (Fig. 4a, b).

Mutagenesis of the F3 binding site on SOCS2 complemented the peptide mutagenesis and confirmed the crystallographic analysis (Fig. 5a and Table 2). Substitution of Leu36, Leu40, Leu81, and Tyr99 with other hydrophobic residues had a significant effect on F3 binding, with L81F and Y99A abrogating binding completely, confirmed by their inability to enhance GHR pY595 binding. L40A and L36A retained the ability to enhance GHR pY595 binding in the presence of F3, indicating there was some residual F3 binding. L82F had a moderate effect on the ability to bind F3 and enhance GHR pY595, and the L40F substitution was accommodated without a significant effect on binding. Mutation of Tyr49 to phenylalanine did not impact GHR pY595 or F3 binding, suggesting that Asp74 is sufficient to maintain the interaction with His6 in F3. Mutation of residues proposed to interact with Lys10 cannot be mutated, as they are all backbone carbonyls. Of the mutations tested, only S78A had a direct effect on GHR binding. In the higher resolution structures of SOCS2 (e.g., 2C9W[6]), the S78 side chain H-bonds with the

peptide backbone, and mutation presumably affects the conformation of the BC loop.

**F3 displays a high degree of selectivity for the SOCS2-SH2 domain.** Several SH2 domain-containing proteins were tested for binding to F3 (CIS, SOCS3, and SH2B). Of these, only the closely related SOCS protein CIS, which shares 45% a.a. identity with SOCS2 within the SH2 domain[38], showed detectable binding by ITC (13.74 μM); albeit considerably weaker than that observed for SOCS2 (1 μM). Notably, F3 binding to CIS still resulted in a fourfold enhancement of pTyr-peptide binding (Supplementary Fig. 5b), suggesting that the exosite phenomenon might extend to other SOCS family SH2 domains and that at least for CIS, an "F3-like" peptide could result in comparable enhancement of pTyr-binding affinity. Sequence alignment indicates that although key SOCS2-ESS residues involved in interacting with F3 differ in CIS, they are largely conserved; for instance, Leu40 in SOCS2 (corresponding to Phe74 in CIS) (Supplementary Fig. 5a, c). Consistent with this, CIS-F74L and SOCS2-L40F had no impact on F3 binding (Table 2 and Supplementary Fig 5b). Sequence alignment of CIS and SOCS2 suggested that the two phenylalanine residues in CIS (Phe74 and Phe116), instead of the two opposing SOCS2 leucines (Leu40 and Leu82), were likely to partially block F3 interaction. However, reverting the CIS phenylalanine residues to leucine also failed to rescue F3 binding (CIS-F74L, F116L; Supplementary Figs. 5b, 10a). Although it is still unclear why CIS binds with lower affinity to F3, it seems likely that differences in multiple residues within both the ESS and β-strand D, collectively result in the reduced affinity (Supplementary Fig. 5c).

**F3 enhances SOCS2-SH2 affinity for pTyr ligands by reducing the rate of dissociation.** Although F3 binding efficiently enhances SOCS2-SH2 domain affinity for pTyr ligands, our structure shows that F3 does not directly contact residues involved in binding phosphotyrosine or residues within the hydrophobic +3 binding pocket. Furthermore, no single SOCS2 residue was identified by mutagenesis, which contributed to both F3 and pTyr-peptide binding, nor did an overlay of the SOCS2/F3 structure with that of SOCS2 bound to a pTyr-peptide (PDB: 6I4X) reveal any global structural differences (Supplementary Fig. 6). Given these observations, we speculated that F3 might function by pre-ordering the BC loop for pTyr engagement. We therefore performed direct binding experiments to measure the on and off rates in the presence and absence of F3. F3 binding caused a significant (~10-fold) reduction in the rate of dissociation ($k_{off}$: 0.245 ± 0.003 to 0.0235 ± 0.001 sec⁻¹ with F3) (Fig. 4c; Supplementary Fig. 10b). Thus, F3 binding appears to stabilize the pTyr-binding pocket, significantly inhibiting dissociation of pTyr ligands, and consequently enhancing affinity.

**F3 enhances SOCS2 inhibition of GH signaling in A549 cells.** Several SOCS2 mutations (including L36A, Y99A, and L81F)

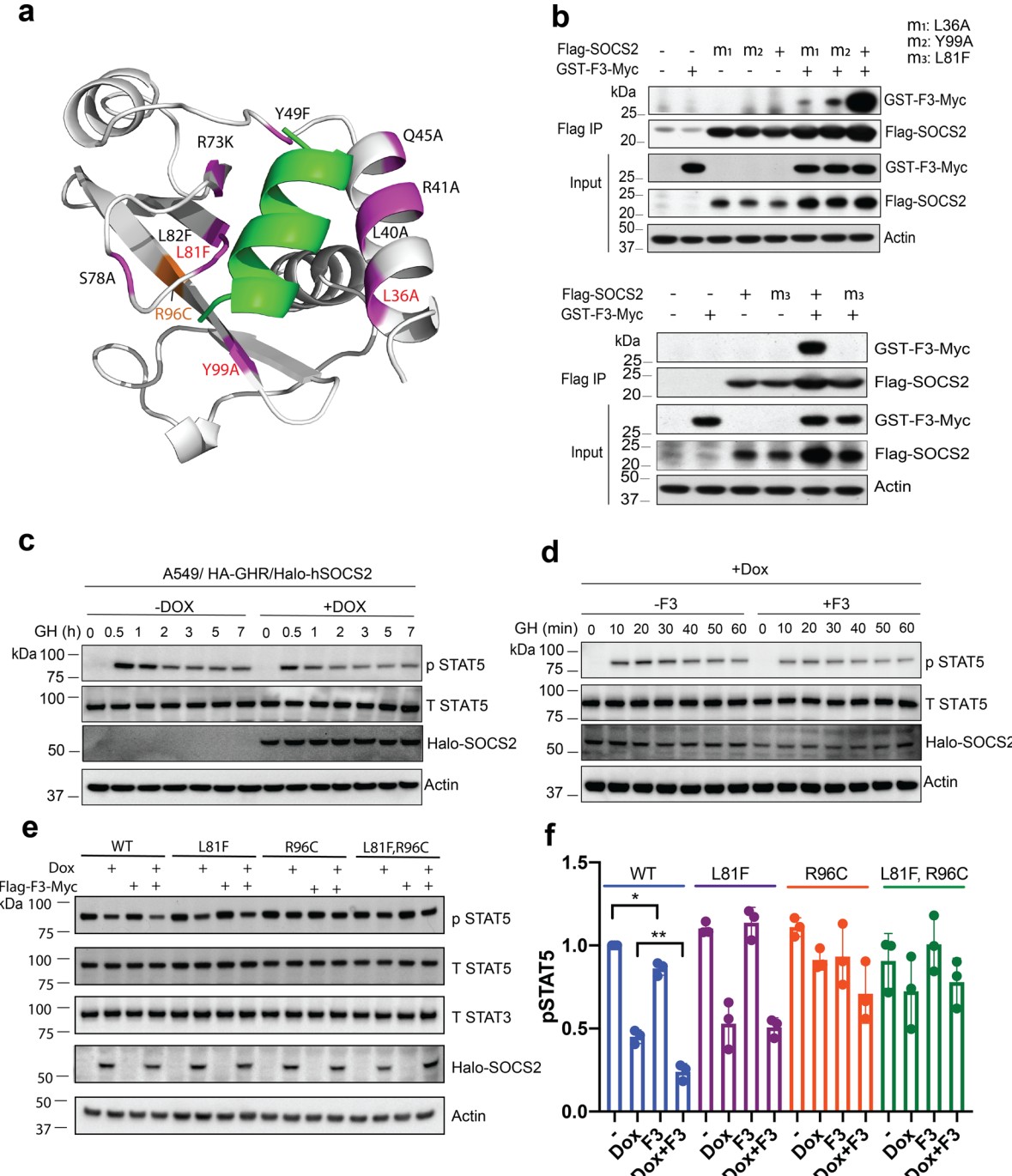

**Fig. 5 F3 binding stabilizes SOCS2 and enhances SOCS2 inhibition of GH signaling. a** F3-bound SOCS2 crystal structure highlighting residues in magenta that were mutated and tested by SPR. Red labels indicate mutations that disrupted F3 binding. Related to Table 2. **b** 293T cells were transfected with constructs for wild-type Flag-SOCS2 or SOCS2 with key F3-contacting residues mutated, with and without GST-F3-Myc expression. Cells were treated with GH and lysed prior to immunoprecipitation of SOCS2 and immunoblotting with anti-Myc antibodies. All three mutations significantly reduced SOCS2 interaction with F3, with SOCS2-L81F resulting in complete loss of F3 interaction. m1: SOCS2-L36A; m2: SOCS2-Y99A, m3: SOCS2-L81F; *IP* immunoprecipitation. Representative of two independent experiments. **c** A549 cells with stable expression of HA-GHR (A549-GHR) were treated overnight with doxycycline (dox) to induce Halo-SOCS2 expression and then stimulated with 50 ng/mL GH for up to 7 h, prior to lysis and analysis by immunoblotting with antibodies to the indicated proteins. Representative of four independent experiments. **d** A549-GHR cells were transfected with Flag-F3-Myc constructs and then treated overnight with doxycycline (dox) to induce Halo-SOCS2 expression. Cells were then treated with 50 ng/mL GH for 0, 10, 20, 30, 40, 50, 60 min, lysed, and analyzed by immunoblotting with antibodies to the indicated proteins. *p* phosphorylated; *T* total (F3). Representative of two independent experiments. **e** A549-GHR cells were transfected with Flag-F3-Myc constructs (F3) prior to overnight dox treatment to induce wild-type (WT) or mutant Halo-SOCS2 proteins. Cells were then treated with 50 ng/mL GH for 30 min, lysed, and analyzed by immunoblotting with antibodies to the indicated proteins. *p* phosphorylated; *T* total. Representative of three independent experiments. **f** Densitometric analysis of pSTAT5 bands relative to WT (no dox, no F3); derived from three independent experiments including data shown in **e** and in Supplementary Fig. 8. Data shown are mean ± S.D. and were analyzed using a paired *t* test. *$p = 0.025$, **$p = 0.0052$. Source data are provided as a Source Data file.

were identified which disrupted F3, but not pTyr binding (Table 2 and Fig. 5a). To validate the SOCS2/F3 interaction in cells, wild-type and mutant Flag-tagged human SOCS2 constructs were generated and expressed in 293T cells, together with a GST-F3-Myc construct. Immunoprecipitation of Flag-SOCS2 and SOCS2 mutants and immunoblotting for the presence of F3, revealed SOCS2-L36A and SOCS2-Y99A still retained weak binding to F3, while SOCS2-L81F completely lost F3 binding (Fig. 5b).

SOCS2 has been reported to inhibit GH signaling and STAT5 phosphorylation[39]. To examine the impact of F3 binding on SOCS2 function, we generated A549 cells with stable expression of HA-GHR and a Halo-tagged SOCS2 fusion protein under a doxycycline (dox)-inducible promoter. Dox-induction of Halo-SOCS2 partially inhibited GH-induced STAT5 phosphorylation, consistent with previous reports[13,39] (Fig. 5c). Consistent with the enhanced affinity conferred by F3 binding, introducing Flag-F3-Myc into the A549/HA-GHR/Halo-hSOCS2 system modestly, but consistently enhanced SOCS2 inhibition of STAT5 phosphorylation (Fig. 5d–f; Supplementary Fig. 8), in particular at 10–30 min GH treatment (Fig. 5d). In our experience the dox-inducible systems are difficult to regulate, perhaps accounting for the relatively modest impact of F3, compared with the increased affinity conferred by F3 as measured by SPR and ITC. Mutation of the F3 binding site on SOCS2 (L81F) prevented F3-mediated enhancement of SOCS2 inhibitory function, whereas mutation of the pTyr-binding site (R96C) reduced SOCS2 inhibition of GH signaling. The SOCS2-L81F, R96C double mutation abrogated both inhibition by SOCS2 and further enhancement by F3 (Fig. 5e, f; Supplementary Fig. 8). Thus, the biophysical enhancement of pTyr-binding affinity conferred by F3 binding to the SOCS2-SH2 exosite translates to enhanced SOCS2 inhibition of GH signaling in cells.

## Discussion

SOCS2 was initially identified as a key negative regulator of GH signaling due to the "gigantic" phenotype of mice lacking Socs2, and was subsequently shown to interact with GHR tyrosine residues via its SH2 domain[6,13,19,25,39]. However, it is the emerging relationship between SOCS2, immunity, and cancer[27,33,38,40] that has generated interest in understanding the role of SOCS2 in cancer etiology. SOCS2 is highly upregulated in androgen-induced prostate cancers and has been shown to promote cancer cell growth, with a reduction in SOCS2 levels inhibiting cell proliferation and xenograft growth[40]. Clinical data also suggest that patients with high SOCS2 expression are more likely to experience biochemical tumor relapse[41]. In the immune system, SOCS2 is an IFNγ signature gene in mononuclear phagocytes infiltrating primary human melanoma, and in mouse models acts to limit adaptive anti-tumoral immunity and DC-based priming of T cells[30]. Understanding how the SOCS2-SH2 domain functions and its full complement of interacting partners will be critical to determine whether SOCS2 might be a suitable target for therapeutic intervention.

The SH2 domain is the prototypical modular signaling domain and binds its cognate pTyr ligands in a defined "two-pocket" paradigm[42]. Using phage display we discovered three non-phosphorylated peptides with a common core motif that bound SOCS2 and had the unique ability to allosterically enhance the affinity of the SH2 domain for phosphorylated peptide ligands and proteins. The peptides therefore led us to identify an exosite on an SH2 domain that positively influences ligand binding. Introducing the exemplar F3 peptide into cells enhanced the ability of SOCS2 to inhibit STAT5 phosphorylation, confirming the increase in affinity translated to a biological outcome in a relevant signaling cascade.

The SOCS2-SH2 domain has the canonical SH2 structural elements (three β-strands flanked by two α-helices) and an additional N-terminal α-helix (ESS) unique to the SOCS family proteins. As in other SOCS proteins, the SOCS2-ESS directly contacts the phosphotyrosine binding loop (BC loop), positioning it for phosphotyrosine binding[6,7,13]. The crystal structure of SOCS2 in complex with F3 (1.68 μM), revealed F3 bound as an α-helix nestled between the ESS and the BC loop, on the opposite side of the pTyr pocket. The 15N-1H HSQC spectrum of the F3 peptide in the absence of SOCS2 displayed typical random-coil chemical shifts (Supplementary Fig. 11), suggesting it does not form a helix in the absence of protein and presumably requires contact with SOCS2 residues to form the secondary structure. This also explains why truncation of the non-conserved F3 residues, which would disrupt helix formation, resulted in the loss of binding to SOCS2 (Table 1).

Surprisingly, neither structural alignment of F3-bound SOCS2 with pTyr-bound SOCS2[35], nor NMR analysis on F3 binding, revealed any major perturbation of SH2 residues that would obviously impact phosphotyrosine binding. On further investigation, we observed a significantly slower dissociation rate for phosphorylated peptides when F3 was bound, which accounts for the increased affinity. Given that the crystal structures in the presence and absence of F3 (within the limits of available resolution) are essentially indistinguishable, but F3 binding significantly decreases the off-rate of bound phosphopeptide, the most likely explanation is that F3 stabilizes the conformation of the BC loop in its bound form and hinders dissociation events.

The F3-driven cooperative increase in binding affinity observed both in vitro and in vivo suggests the SOCS2-SH2 domain may have additional non-canonical interactions that we are unaware of. There is a precedent within the SOCS family, given the SOCS1 and SOCS3-SH2 domains interact with a GQM motif on the JAK1, JAK2, and TYK2 kinase domains; however, this interaction interface is quite distinct from the F3 exosite and does not impact on binding to phosphorylated ligands[9,43,44]. The SOCS2-SH2 domain has the capacity to bind two phosphopeptides simultaneously, with one peptide occupying the canonical site and the second peptide binding in an anti-parallel direction[35]. Again, the non-canonical site does not overlap with the F3 exosite, but this is further evidence that there is still more to understand for the SOCS2 inhibitory signaling cascades.

Protein interaction with the SOCS2 exosite would provide an additional level of biological specificity, further to that provided by the phosphotyrosine motif, and has the potential to greatly expand the range of SOCS2-SH2 targets. This possibility led us to search for endogenous proteins which might occupy the SOCS2 exosite. The central sequence motif (including the tolerated residue changes) was used to search the NCBI human protein databases and although a number of candidate proteins were identified, peptides derived from these proteins did not interact with SOCS2 (Supplementary Table 3). This list is by no means exhaustive, particularly when an "F3-like" ligand may be linear, an α-helix or non-contiguous. It is also possible that another, non-homologous sequence may exist that binds to this site, which may not have been present in the phage display library (or was not able to form an alpha-helix), and other approaches, such as proteomic analysis of SOCS2 complexes, may prove more fruitful.

The SOCS2 exosite could potentially be exploited to enhance SOCS2 suppression of the inflammatory disease. In mice, loss of SOCS2 biases towards CD4+ Th2 differentiation, suggesting patients with allergic diseases such as atopic dermatitis and asthma may benefit from enhanced SOCS2 expression or activity[27]. In cancer patients, SOCS2 expression is associated with both a better and worse prognosis depending on the cancer type. For example, high SOCS2 levels correlate with increased survival

in patients with hepatocellular carcinoma (HCC)[45,46] and invasive breast cancer[47], whereas in metastatic melanoma patients, SOCS2 is a key component of an IFNγ-signature associated with increased survival[30]. This apparent dichotomy may relate to the complex role of inflammatory signaling, which can both drive cancer growth and influence anti-cancer immunity. An F3-mimick that enhanced SOCS2 activity has the potential to reduce JAK/STAT and NF-κB-driven pulmonary inflammation, and in particular, may benefit cancer patients with HCC or prostate cancer.

The identification of the F3 exosite suggests SOCS2-SH2 binding affinities based on one-site interactions may be substantially underestimated, particularly if proteins in the same complex occupy the canonical pTyr site and exosite simultaneously. How widespread the exosite phenomenon might be remains unclear. We have shown that the CIS-SH2 domain, which shares the highest sequence similarity with SOCS2, also contains an exosite that modulates CIS-SH2 binding to phosphorylated ligands (albeit to a lesser extent than SOCS2), indicating this phenomenon extends to at least one related SH2 domain. It is a testament to the complexities of biology that a domain discovered in 1986 with a historic and well-defined paradigm for ligand interaction[48], can still present us with some surprises. Future studies will no doubt reveal whether our findings translate to other SOCS family members and/or to the greater SH2 domain family.

## Methods

**Cloning and purification of SOCS proteins.** Full-length human SOCS2[1–198], SOCS2 lacking the N-terminal region (SOCS2[32–198]), SOCS2-SH2 lacking the N-terminal region, and the SOCS box (SOCS2-SH2[32–159]) and human CIS lacking both the N-terminal region and PEST motif (Δ174–202) (CIS[66–258]), were cloned into pGEX-4T. Human Elongin B (ELOB; residues 1–118) and Elongin C (ELOC; residues 17–112) were co-cloned into pACYCDuet. Constructs encoding GST-SOCS2 or GST-CIS, were co-expressed with Elongin BC in E. coli BL21 (R3) and induced by 1 mM isopropyl β-d-1-thiogalactopyranoside (IPTG) at 18 °C for 15 h. A full list of DNA constructs used are provided in Supplementary Table 4. 1 L S-broth cultures were collected by centrifugation and resuspended in phosphate-buffered saline (PBS) containing 5 mM β-mercaptoethanol, 5 mM phenyl phosphate, 1 mM phenylmethylsulfonyl fluoride, 20 Units DNAse, and 20 mg lysozyme, and lysed by sonication. Supernatants were collected by centrifugation at 50,000 × g for 20 min and purified by affinity chromatography using Glutathione Sepharose resin (GE Healthcare Life Sciences). The resin was washed with PBS containing 5 mM β-mercaptoethanol and 5 mM phenyl phosphate, prior to Tobacco Etch virus (TEV) protease cleavage to remove GST. The trimeric SOCS2/BC complex was finally purified by size-exclusion chromatography on a Superdex 200 16/600 column (GE Healthcare Life Sciences) in 25 mM 4-(2-hydroxyethyl)-1-piperazineethanesulfonic acid (HEPES), pH 7.5, 150 mM NaCl, 5 mM phenyl phosphate, and 2 mM tris(2-carboxyethyl)phosphine (TCEP). SOCS2/BC complexes containing mutant SOCS[32–198] proteins were produced and purified as described above.

**Cullin5 purification.** Expression and purification of the GST-Cullin5 N-terminal domain (Cul5[1–384]) have been described previously[49]. In brief, protein production was induced using IPTG at 25 °C overnight. TEV cleaved protein was run on a Superdex 200 16/60 size-exclusion column equilibrated in Tris-buffered saline containing 5 mM β-mercaptoethanol.

**Phage display.** Phage display selections were performed against the GST-SOCS2[32–198]-EloB/C complex essentially as described[50,51], using a randomized 16-mer peptide library fused to the gene-8 major coat protein of M13 phage. Two sequential negative selection steps on BSA and GST were used to remove non-specific binders. Phage were then cycled through five rounds of enrichment for binding to the GST-SOCS2[32–198]-EloB/C complex. Specific binding of the phage was confirmed by enzyme-linked immunosorbent assays (ELISA) and clones displaying binding to GST-SOCS2, but not to GST or BSA, were subjected to Sanger DNA sequencing. Competitive phage-ELISAs were used to estimate the affinity of peptides bound to the GST-SOCS2[32–198]-EloB/C complex as described[52].

**Isothermal titration calorimetry.** All ITC titrations were performed at 298 K using a Microcal ITC200 (GE Healthcare Life Sciences), essentially as described[10]. Lyophilized peptides were obtained from Genscript and resuspended in running buffer (25 mM Hepes pH 7.5, 150 mM NaCl, 2 mM TCEP). Typically, 12 × 3.25 μL

injections of a 240 μM peptide solution were titrated into a 30 μM solution of SOCS-EloB/C. An initial injection of 0.4 μL peptide solution was performed and removed from the analysis. To determine the pTyr-peptide affinities for the SOCS2/F3 complex, 25 μM SOCS2[32–198]-EloB/C was pre-incubated with 60 μM F3. Data were analyzed using Microcal Origin 7.0. The binding curves fitted a single-site binding mode and KD values were determined from duplicate experiments. ITC curves and parameters are shown in Supplementary Fig. 9; and Supplementary Tables 1 and 2.

**Thermal shift analysis.** In all, 13 μM of purified SOCS2[32–159] (SOCS2-SH2 domain alone) was incubated with either 50 μM GHR pY595 peptide, 50 μM F3 peptide, or both peptides in HEPES-buffered saline for 5 min before addition of Sypro-Orange (Sigma). Samples were then dispensed in triplicate into a white 384-well PCR plate (Thermo Scientific). Sample heating was performed on a C1000 Thermal Cycler (Biorad) with a programmed temperature ramp rate of 1 °C/min beginning at 25 °C and ending at 95 °C. Sample fluorescence was read in the "FRET" channel of a CFX384 Real-Time System (Biorad). TM was obtained by performing a Boltzmann sigmoidal non-linear fit of normalized data that has been truncated after 100% fluorescence was reached. Data were analyzed in Prism 9 (version 9.0.0) for macOS.

**Mutagenesis of SOCS2 and CIS.** SOCS2 and CIS mutations were introduced using the QuikChange Lightning Site-Directed Mutagenesis Kit (Agilent Technologies). Primers for mutagenesis were purchased from Integrated DNA Technologies (Singapore). Primer sequences are listed in Supplementary Table 6.

**Surface plasmon resonance measurement of SOCS protein:peptide-binding affinities.** Experiments were performed on a BIACORE 4000 (GE Healthcare) at 18 °C in HBS-ET (10 mM HEPES pH 7.4, 150 mM NaCl, 3.4 mM EDTA, 0.005% Tween 20). For measurement of GHR pY595 peptide binding, biotin-GSGS-GHR pY595 peptide was immobilized to a Streptavidin-coated SA chip (GE Healthcare). A reference flow cell was prepared using immobilized biotin-GHR Y595 (no phosphorylation). 100 nM SOCS proteins were pre-incubated with titrations of phosphorylated competitor peptide (10 μM, 3.3 μM, 1.1 μM, 0.3 μM, 0.1 μM) in HBS-ET and flowed through the chip. Binding responses were measured using multi-cycle kinetics, with 240 s injection and 120 s disassociation at a flow rate of 30 μL/min. 50 mM NaOH, 1 M NaCl was used to regenerate the chip surface. F3 enhancement of GHR pY595 peptide affinities was measured using a fixed concentration of F3 (10 μM) and SOCS2 or CIS (100 nM) with titrations of GHR pY595 peptide.

For measurement of F3 peptide binding, biotin-GSGS-F3 peptide was immobilized to a Streptavidin-coated SA chip, with biotin immobilized in a reference flow cell. 1 μM SOCS proteins were pre-incubated with titrations of F3 competitor peptide (10 μM, 3.3 μM, 1.1 μM, 0.3 μM, 0.1 μM) in HBS-ET plus 5 mM phenyl phosphate and flowed through the chip using multi-cycle kinetics. Values were derived from three independent experiments (mean values ± S.D.). Raw data was analyzed using the Biacore evaluation software and Prism 8 (version 8.0.2). SOCS protein with no competitor phosphopeptide was used to determine maximal binding. The relative binding values were plotted against the Log10 of competitor peptide and a steady-state analysis was performed to derive IC50.

**Cell culture and transient transfection.** Human Embryonic Kidney cells (HEK293T; RRID:CVCL_0063), human lung adenocarcinoma A549 cells (RRID:CVCL_0023), and human HCC HepG2 cells (RRID:CVCL_0027) were cultured in Dulbecco's Modified Eagle's Medium supplemented with 100 U/mL penicillin, 0.1 ng/ml streptomycin and 10% fetal bovine serum (Thermo). All cell lines were tested as mycoplasma-free. HEK293T and A549 cells stably expressing Hemagglutinin-tagged GHR were generated by lentiviral transfection and selection in puromycin (293T-GHR and A549-GHR, respectively). HEK293T cells were transiently transfected with pEFBOS[53] constructs expressing Human Flag-SOCS2 and GST-F3-Myc, using FuGene6 (Promega) according to the manufacturer's instructions. A549 cells were transiently transfected with Flag-F3-Myc or GST-F3-Myc (pEFBOS) constructs using Lipofectamine 2000 (Thermo Fisher) according to the manufacturer's instructions. HEK293T and A549 cells were lysed 48 h post transfection for further analysis.

**Stable transfection of HEK293T and A549 cells.** To generate A549 cells expressing GHR together with doxycycline-inducible Halo-tagged SOCS2 or SOCS2 mutants, cells were first stably transfected with an HA-tagged human GHR construct (pQCXP) using retrovirus infection, and then with Halo-tagged human SOCS2 and SOCS2 mutant constructs (Tre3g) using lentivirus infection. HEK293T cells expressing mRuby3 and Halo-SOCS2 were generated using lentivirus infection. The virus was produced by transient co-transfection of Packaging and envelop vectors in HEK293T cells using Lipofectamine 2000 (Thermo Fisher). Packaging and envelop constructs are listed in Supplementary Table 4. Infections were performed as described previously[54]. A549 cells transduced to express HA-GHR were selected with 2.5 μg/mL puromycin for 7 days. HA-GHR expression was confirmed by immunoblotting and cells were transduced with Halo-tagged SOCS2 and mutant SOCS2 constructs. Halo-SOCS2 expression was induced with 1 μg/mL

doxycycline overnight, together with 10 nM Janelia Fluor (JF)646 HaloTag (Promega; GA1120) to label Halo-SOCS2 protein. JF646 positive cells with comparable expression were sorted using a BD FACSAria III cell sorter, prior to expansion in culture. An example gating strategy is shown in Supplementary Fig. 13.

**Affinity enrichment and immunoblotting.** HEK293T cells and A549 cells were lysed in NP-40 lysis buffer [1% v/v NP-40, 50 mM HEPES, pH 7.4, 150 mM NaCl, 1 mM EDTA, 10% glycerol, 1 mM PMSF, 1 mM $Na_3VO_4$, 1 mM NaF and protease inhibitors (Complete Cocktail tablets, Roche)][12]. Cell lysates were clarified by centrifugation at $13,000 \times g$ for 15 min at 4 °C. For affinity enrichment, the supernatant was pre-cleared with protein-A Sepharose (Sigma) prior to affinity enrichment with anti-Flag M2 affinity agarose gel (Sigma) or Glutathione Sepharose resin (GE Healthcare) for 2.5 h at 4 °C. Recovered proteins were separated by sodium dodecyl sulfate-polyacrylamide gel electrophoresis and electrophoretically transferred to nitrocellulose membranes (Amersham). Membranes were blocked overnight in 5% w/v BSA and incubated with primary antibody for 2 h. Anti-phosphotyrosine antibody (ab179530; 1:2000) was obtained from Millipore. Anti-STAT3 antibody (4904; 1:2000), anti-Myc antibody (3946; 1:2000), anti-STAT5 (94205; 1:2000), and anti-phospho-STAT5 (9359; 1:2000) were purchased from Cell Signaling. Anti-phospho-JAK2 antibody was obtained from Millipore (07-606; 1:1000) and anti-JAK2 antibody from Santa Cruz (sc-390539; 1:1000). Anti-actin-HRP antibody (C4) was obtained from Santa Cruz (sc-47778 HRP; 1:1000). Anti-Halo antibody (G921A; 1:2000) was obtained from Promega and anti-HA antibody (12158167001; 1:1000) from Sigma Aldrich. Rat anti-Flag antibody was a kind gift from Prof. D. Huang & Dr. L. O'Reilly (Walter and Eliza Hall Institute). Antibody binding was visualized with peroxidase-conjugated goat anti-rat immunoglobulin (Southern Biotech; 3010-05; 1:10000), sheep anti-rabbit immunoglobulin (Southern Biotech; 4010-05; 1:15000), or sheep anti-mouse immunoglobulin (GE Healthcare; NA931-1ML; 1:10000) and the enhanced chemiluminescence (ECL) system (Amersham or Millipore).

**FLIM-FRET interaction assay in live cells.** HEK293T cells stably expressing Halo-SOCS2 (dox-inducible) and/or mRuby3-F3 (constitutive) were induced with doxycycline (2 μg/mL) and labeled with Halo ligand JF503 (50 nM) overnight on Ibidi chambered coverslips. Live cells were imaged on a Leica SP8 confocal microscope with a FALCON fluorescence lifetime imaging detector (80 MHz laser repetition rate, IRF generated from reflected light). Fluorescence lifetime data was analyzed using FLIMfit (v4.12.149). Mono-exponential fluorescence lifetimes were fitted pixel-wise (11 × 11 bin) for the FLIM images, and fitted image-wise for the plots and statistical analysis.

**X-ray crystallography.** Purified $SOCS2^{KKQ}BC$ was prepared in 25 mM HEPES (pH 7.5), 250 mM NaCl, 10 mM DTT and 5 mM phenyl phosphate. Three times molar excess of truncated F3 peptide (ALQHLMDKWMAM) was incubated with $SOCS2^{KKQ}BC$. The sample was concentrated to 15 mg/mL and sodium cacodylate pH 7.2 was added to a final concentration of 0.1 M before crystallisation[35]. Diffraction quality crystals grew as oval plates and were obtained in 2 M ammonium sulfate, 0.1 M Bis-tris chloride pH 6.5. Crystals were transferred to a cryoprotectant solution consisting of 2 M ammonium sulfate, 0.1 M Bis-tris chloride pH 6.5, 25% glycerol, and flash-frozen in liquid nitrogen. Diffraction data were collected on beamline MX2 at the Australian Synchrotron[55] using a wavelength of 0.9373 Å. Data were integrated to 3.1 Å using XDS[56].

The structure was determined by molecular replacement with PHASER as implemented in PHENIX using the SOCS2/BC structure from PDB ID 2C9W[6]. Serial rounds of refinement and model building were performed using PHENIX[57] and COOT[58], respectively. Refinement converged to $R_{work}$ of 0.2271 and $R_{free}$ of 0.2667. The final refined model had 96% residues in the favored region and 3% residues in the allowed region of the Ramachandran plot. See Supplementary Table 5 for data collection and refinement statistics and Supplementary Fig. 12 for a portion of the electron density map.

**NMR spectroscopy.** $hsSOCS2\text{-}SH2^{32-159 (\Delta SOCS2box)}$ was cloned in pGEX-4T as a GST fusion protein. Protein expression was induced with 1 mM IPTG at 25 °C for 4–6 h in BL21 (DE3) cells in M9 minimal media containing 1 g/L $^{15}N$ ammonium chloride and 4 g/L $^{13}C$-glucose as the sole nitrogen and carbon source, respectively. The protein was purified by affinity chromatography and size-exclusion following protocols described above, except with an increase in NaCl concentration to 500 mM, and the addition of 20 units RNase during the lysis step. For NMR studies, the protein was concentrated to 70 μM and buffer-exchanged into 20 mM sodium phosphate (pH 6.8), 50 mM NaCl, 2 mM TCEP. Protein backbone assignments were performed by analyzing HNCA, HNCACB, and $^{15}N$-NOESYHSQC spectra recorded at 25 °C on a Bruker Avance 600 MHz spectrometer equipped with a triple-resonance cryoprobe. Spectra were processed in TopSpin and analyzed in CARA. NMR analyses of peptide-bound and F3-bound SOCS2-SH2 were performed by collecting $^1H\text{-}^{15}N$ SOFAST HMQC spectra with 70 μM in the presence of a twofold excess of GHR pY595 peptide and twofold excess of F3 peptide. CSPs in the $^1H$ and $^{15}N$ dimensions were calculated following the previously described methodology[37]. Assignments have been deposited with the Biological Magnetic Resonance Bank (accession no. 50869 and 50868).

**Reporting summary.** Further information on research design is available in the Nature Research Reporting Summary linked to this article.

## Data availability

The coordinates and structure factors of human SOCS2 in complex with elongins B and C and F3 peptide generated in this study have been deposited in the RSCB Protein Data Bank (PDB) (http://www.rcsb.org/pdb) under accession code 7M6T. The NMR assignments generated in this study have been deposited with the Biological Magnetic Resonance Bank (https://bmrb.io/) under accession codes 50868 (SOCS2-SH2 bound to GHRpY peptide), which is available at, and 50869 (SOCS2-SH2 bound to GHRpY peptide and F3 peptide), which is available at. Previously published crystal structures used in this study are available from the PDB under accession codes 2C9W, 5BO4, and 6I4X. Source data are provided with this paper.

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

## Acknowledgements

J.J.B was supported by an Australian Government National Health and Medical Research Council (NHMRC) Research Fellowship (1121755). N.K. was supported by an Australian Government Research Training Program Scholarship. K.L. was supported by a Melbourne Research Scholarship (University of Melbourne). This work was supported in part by NHMRC Project Grant (1124784) and through Victorian State Government Operational Infrastructure Support and the Australian Government NHMRC Independent Research Institutes Infrastructure Support Scheme (IRIISS). This research was undertaken in part using the MX2 beamline at the Australian Synchrotron, part of ANSTO, and made use of the Australian Cancer Research Foundation (ACRF) detector.

## Author contributions

E.M.L. designed and analyzed experiments. Generated protein and performed ITC experiments. Discovered the effect of F3 on phosphotyrosine binding. Supervised protein production, ITC, and SPR experiments. K.L. generated protein for SPR and crystallization experiments. Performed all competitive SPR. Performed and analyzed association/dissociation experiments. Generated all SOCS2 mutant constructs. Generated cell lines. Performed signaling experiments and affinity enrichment experiments. G.V. performed and analyzed phage display experiments and validated peptide hits. C.T. produced labeled protein. Performed and analyzed NMR experiments and thermal shift analysis. F.D. generated GHR and Halo-SOCS2-expressing cell lines and performed initial experiments. Designed Flag-F3-Myc construct. C.H. performed and analyzed FLIM-FRET experiments. D.C. generated protein, performed GST affinity enrichment experiments and ITC experiments. N.K. generated Halo-SOCS2 construct. R.F. Provided expression vectors and advice. A.J.B. provided GHR expression vector, GH, and advice. S.S.L. supervised phage display experiments and interpreted data. S.S.S. provided OPAL and permutation arrays used to investigate SOCS2-binding preferences (not included in the final manuscript). J.J.B. Supervised ITC, SPR, and NMR experiments and interpreted data. Conceived and interpreted association/dissociation experiments. NJK: Solved the SOCS2/F3 crystal structure. S.E.N. and S.S.l. conceived the initial investigation using phage display. S.E.N. and N.J.K. oversaw and led the study and co-wrote the paper with K.L. All authors reviewed the manuscript.

## Competing interests

S.E.N., J.J.B., and N.J.K. receive research funding from a pharmaceutical partner. The funders had no role in the content or writing of the manuscript. The remaining authors declare no competing interests.
