## [Peer Review File · Nature Communications]

REVIEWER COMMENTS

Reviewer #1 (Remarks to the Author):

This study by Linossi et al. identifies and characterizes a 16-residue peptide that is capable of binding to the SOCS2 SH2 domain and enhancing its phosphopeptide binding properties. The authors used a variety of biochemical and biophysical/structural experiments to show that this peptide (F3) binds to the face of the SH2 domain opposite to the phosphopeptide (pTyr) binding surface, and that binding of this peptide, as an alpha helix, markedly decreases the off-rate of the bound phosphopeptide, leading to higher affinity binding than in the absence of F3. Moreover, expression of F3 in cells led to increased SOCS2 inhibition of signaling by growth hormone receptor (GHR).

This is an interesting and potentially important study, with solid experimental data, showing that the “back side” of the SOCS2 SH2 domain, and possibly of other SH2 domains, could be a site for allosteric regulation. Although the authors show that F3, when co-expressed in cells, can potentiate SOCS2 downregulation of GHR signaling, i.e., this is not merely an in vitro oddity, they were unable to identify F3-related sequences in endogenous proteins that could bind to the SOCS2 SH2 domain in vitro. Therefore, the in vivo relevance of the F3-SOCS2 SH2 interaction is in question. Nevertheless, this discovery should spark further investigation into putative endogenous potentiators of SH2 domains and the use of exogenous F3-like peptides as possible therapeutics or cellular tools.

Reviewer #2 (Remarks to the Author):

This work investigates the binding of a phage display-derived peptide (F3) to the SH2 domain of SOCS2 protein. The peptide is found to bind to a region of the SH2 domain which is removed from the canonical phosphotyrosine (pY) binding site. In vitro binding data show that the presence of the F3 peptide enhances binding of the SH2 domain to peptides corresponding to the cognate pY binding site on growth hormone receptor. Cell-based assays are used to demonstrate the effects of F3 on SOCS2 binding in a more ‘physiologically relevant’ context.

The work is well presented and the structural and biophysical characterisation of binding between F3 and SOCS2 SH2 domain is convincing (with some caveats – see below). However, there are several points that require attention.

- 1) There is very little detail on the phage display outputs. There is no information on other sequences that might have been observed, the degeneracy of the individual screens, enrichment

through the five rounds etc. It is important for the reader to understand how stringent the phage display screening was and what control and selection criteria were imposed. The phage display was performed in the absence of the first 31 amino acids of SOCS2, some comment on this should be made in the Results section.

2) Biophysical characterisation with ITC requires inclusion of the errors of the fits for KD.

3) The raw SPR data and data fitting are required (e.g. Fig 3c). For example, the fitting of the fast off rates to raw data attempted there is usually difficult and often unconvincing. This should be included in the Supplementary Data.

4) The experiment shown in 2c is problematic and does not appear to be adequately controlled across the conditions adopted. It would be more convincing if western blotting could be adopted to confirm the identify important proteins observed.

5) The experiment shown in 2f appears to show negligible binding of SOCS2 to GHR in the absence of F3. This infers that F3 is required for interaction which we know if not the case. Perhaps densitometric quantification and analysis of more than one blot would clarify this.

6) The cell-derived assays shown on the blots in Fig 5 are potentially difficult to interpret and some comment in the text on the weaknesses of this approach are required. The blots of pSTAT (5d) in particular are not entirely convincing. The problem might be due to the observation that the presence of F3 appears to increase affinity of the SOCS2 SH2 domain by approximately an order of magnitude. Within the context of the cell concentration fluctuations imposed as a result of dox-induced expression systems are difficult to control and can be almost impossible to predict. Therefore, it is hard to attribute changes in the pSTAT to F3 without quantification of the F3 expression levels.

Overall the observation of the binding of a peptide to an exosite on SOCS2 SH2 is of interest and the characterisation of this is of general relevance as more evidence of non-canonical binding associated with SH2 domains is reported. Much of the analysis of the role of the F3 sequence in cells is over-interpreted and it is not really clear what the relevance of this is, particularly in the light of the fact that the sequence is not attributable to a known cellular ligand, or there is no proven therapeutic application of the use of F3.

Reviewer #3 (Remarks to the Author):

Linossi and colleagues have identified and targeted a novel allosteric binding site in the SH2 domain of SOCS2. The SOCS family proteins are key regulators of cytokine signaling with central roles in physiological signaling and in recent years critical functions were also identified in pathophysiological signaling in cancer and immunity. From a phage display library, the authors identified non-phosphorylated peptides that bound SOCS2 with low micromolar affinity. Excitingly, peptide F3 did not compete with phosphotyrosine (pY) ligand binding to the SOCS2 SH2 domain, but instead strongly enhanced pY peptide binding both in vitro and in pull-down assays from cells. Subsequent co-crystallization of F3 with a SOCS2-Elongin C-Elongin B complex showed an alpha helical conformation of F3 and a well-defined binding site, which was confirmed by extensive mutagenesis and NMR chemical shift perturbation mapping. The binding site is located between the N-terminal ESS helix (unique to the SOCS family) and the BC loop (adjacent to the pY binding pocket). SPR experiments then revealed the kinetic mechanism for enhanced binding to pY by F3 (mainly strongly reduced koff). Finally, the authors demonstrate enhanced SOCS2-dependent inhibition of growth hormone signaling in cells by expressing F3.

This is an excellent piece of work that clearly demonstrates the presence of a previously unrecognized exosite that strongly regulates substrate recognition of the SOCS2 SH2 domain. This new exosite has not been previously identified despite more than a dozen crystal structures of SOCS family members. The study uses a wealth of quantitative biochemical and structural biology methods and confirmed results in cellular models. Experiments are well controlled and clearly presented and described. The findings are entirely novel and important evidence is presented that besides SOCS2, also the SOCS family member CISH could be regulated by ligands to this novel exosite. The findings are of great importance, as they indicate that substrate recognition by SOCS proteins could be strongly modulated by endogenous proteins that bind the novel exosite. Secondly, engineered ligands, such as F3, could be used therapeutically to enhance SOCS2-mediated inhibition of activated cytokine receptor signaling in various disease settings.

Minor points to be addressed:

1. Fig 1b: The ITC data for binding of F3, C4 and E11 should be shown in SI.
2. Table 1 reports binding affinities of mutants that were derived by either ITC or SPR. The Kd for F3 wt should be shown for both SPR and ITC to see if comparable values are obtained so that one can compare Kds of mutants obtained by the two different methods.
3. line 107/108: ... truncations abrogate binding ... : Is the F3 peptide alone (not bound to SOCS2) alpha helical? And could the truncation result in loss or decreased degree of alpha helicity? This could be easily tested by far-UV CD spectroscopy.
4. line 118/119: ... F3 ... did not compete with pY peptide from GHR: Where is this data shown? no reference to Fig. or SI Fig?
5. line 171: typo in PDB entry: 6I4X and not 614X.
6. line 224-256: It would be interesting to add thermal shift data to determine how much F3 binding may increase the melting temperature of SOCS2.

7. Figure 5d and 5e lack anti-FLAG or anti-Myc immunoblot showing expression of the FLAG-F3-Myc construct. These should be added.

8. The Discussion section rather vaguely speculated on SOCS2 as a possible therapeutic target. Some sentences should be added for which diseases and under which circumstances this might be useful. In addition, if and how F3 could be exploited therapeutically should be discussed: If F3 is alpha-helical when not bound to SOCS2, helix stapling and peptide delivery to cells could be attempted in a follow-up study.

RESPONSE TO REVIEWER COMMENTS

Reviewer #1 (Remarks to the Author):

This study by Linossi et al. identifies and characterizes a 16-residue peptide that is capable of binding to the SOCS2 SH2 domain and enhancing its phosphopeptide binding properties. The authors used a variety of biochemical and biophysical/structural experiments to show that this peptide (F3) binds to the face of the SH2 domain opposite to the phosphopeptide (pTyr) binding surface, and that binding of this peptide, as an alpha helix, markedly decreases the off-rate of the bound phosphopeptide, leading to higher affinity binding than in the absence of F3. Moreover, expression of F3 in cells led to increased SOCS2 inhibition of signaling by growth hormone receptor (GHR).

This is an interesting and potentially important study, with solid experimental data, showing that the “back side” of the SOCS2 SH2 domain, and possibly of other SH2 domains, could be a site for allosteric regulation. Although the authors show that F3, when co-expressed in cells, can potentiate SOCS2 downregulation of GHR signaling, i.e., this is not merely an in vitro oddity, they were unable to identify F3-related sequences in endogenous proteins that could bind to the SOCS2 SH2 domain in vitro. Therefore, the in vivo relevance of the F3-SOCS2 SH2 interaction is in question. Nevertheless, this discovery should spark further investigation into putative endogenous potentiators of SH2 domains and the use of exogenous F3-like peptides as possible therapeutics or cellular tools.

While we were unable to identify an F3-like sequence by database searching, another non-homologous sequence may exist that binds to this site, which was not present in the phage display library (or was not able to form an alpha helix). The discussion around this point has been expanded (Discussion p16).

Reviewer #2 (Remarks to the Author):

This work investigates the binding of a phage display-derived peptide (F3) to the SH2 domain of SOCS2 protein. The peptide is found to bind to a region of the SH2 domain which is removed from the canonical phosphotyrosine (pY) binding site. In vitro binding data show that the presence of the F3 peptide enhances binding of the SH2 domain to peptides corresponding to the cognate pY binding site on growth hormone receptor. Cell-based assays are used to demonstrate the effects of F3 on SOCS2 binding in a more ‘physiologically relevant’ context. The work is well presented and the structural and biophysical characterisation of binding between F3 and SOCS2 SH2 domain is convincing (with some caveats – see below). However, there are several points that require attention.

1) There is very little detail on the phage display outputs. There is no information on other sequences that might have been observed, the degeneracy of the individual screens, enrichment through the five rounds etc. It is important for the reader to understand how stringent the phage display screening was and what control and selection criteria were imposed.

We have expanded our description of the phage display panning in the first part of the Results section (p6). In addition to the three homologous phage peptides described in this manuscript, only one other peptide was identified that bound to the SOCS2-SOCS box (not to the SH2 domain). We have not included the data here as we hope to characterise this interaction further for a future publication. We trust that this satisfies the reviewer’s concerns.

The phage display was performed in the absence of the first 31 amino acids of SOCS2, some comment on this should be made in the Results section.

We have now made it clear in the text that SOCS2 lacking the N-terminal region was used in the phage panning, using the descriptor GST-SOCS2³²⁻¹⁹⁸-EloB/C. Note the truncated SOCS2 complex was used as it is more stable than the full-length SOCS2 construct and protein needed to be shipped to Toronto from Melbourne for phage panning. We have shown by SPR that F3 binds to full-length SOCS2 with similar affinity to the truncated SOCS2 protein and similarly enhances pTyr peptide binding (discussed in Results section, p8, line 144, with data shown in Supplementary Figure 2a).

2) Biophysical characterisation with ITC requires inclusion of the errors of the fits for KD.

ITC errors of the fits for KD have now been included in Supplementary Table 2.

3) The raw SPR data and data fitting are required (e.g. Fig 3c). For example, the fitting of the fast off rates to raw data attempted there is usually difficult and often unconvincing. This should be included in the Supplementary Data.

Representative examples of the transformed SPR data have been included in Supplementary data (Supplementary Fig. 10a) with the means +/- SD included in Tables 1 & 2 and Supplementary Fig. 2a. The on/off fitting (rather than data curves) in Fig. 4c were included in error and the fitting curves have now been replaced with the appropriate SPR data curves. Data

curves and fitting are now shown for each association/dissociation experiment in Supplementary Fig. 10b. If accepted, all raw SPR data will be provided as open source data in an excel file format.

4) The experiment shown in 2c is problematic and does not appear to be adequately controlled across the conditions adopted. It would be more convincing if western blotting could be adopted to confirm the identify important proteins observed.

Figure 2c shows F3-enhanced enrichment of tyrosine phosphorylated proteins by GST-SOCS2²⁸⁸⁻¹⁹⁸/BC and confirms the impact of F3 on SH2 binding to tyrosine phosphorylated proteins (rather than peptides) in a complex cellular lysate. In this experiment the same cellular lysate is split into the different GST enrichment conditions. GST was used as a negative control and competition with a phospho-Tyr peptide was used to confirm proteins were enriched via canonical pTyr:SOCS2-SH2 interactions. The GST-input is given in the right-hand side Coomassie stained panel. Re-probing with GST is not feasible by immunoblotting as the GST signal would be too strong. It is unclear what additional controls could have been incorporated. Western blotting for specific interactors JAK2 and GHR is shown in Figure 2f.

5) The experiment shown in 2f appears to show negligible binding of SOCS2 to GHR in the absence of F3. This infers that F3 is required for interaction which we know if not the case. Perhaps densitometric quantification and analysis of more than one blot would clarify this.

We appreciate the apparent inconsistency highlighted by the reviewer. We do indeed believe that SOCS2 interaction with GHR occurs in the absence of F3, and that the lack of detection results from a technical limitation in detecting small amounts of protein with this antibody. The SOCS2:GHR interaction was not detected in 2/3 experiments in the absence of F3. Densitometry would not assist in this instance, given that no band was visible above background in 2 experiments (in the absence of F3), even with long exposure times. All 3 experiments are shown in the figure below, with a representative figure (experiment 2) included as Figure 2f.

6) The cell-derived assays shown on the blots in Fig 5 are potentially difficult to interpret and some comment in the text on the weaknesses of this approach are required. The blots of pSTAT (5d) in particular are not entirely convincing. The problem might be due to the observation that the presence of F3 appears to increase affinity of the SOCS2 SH2 domain by approximately an order of magnitude. Within the context of the cell concentration fluctuations imposed as a result of dox-induced expression systems are difficult to control and can be almost impossible to predict. Therefore, it is hard to attribute changes in the pSTAT to F3 without quantification of the F3 expression levels.

We agree that there are aspects to these experiments that are technically challenging, including as commented on by the reviewer, that the dox-inducible systems are difficult to regulate (i.e. in our experience they are either on or off, and titration of dox doesn't result in a corresponding titration of protein). Detection of low molecular weight proteins by immunoblotting is a common issue in the field (i.e. Suzuki et al., Anal Biochem., 2008; <https://www.ptglab.com/news/blog/tech-tips-in-search-of-low-molecular-weight-proteins/>). We have spent considerable effort trying to detect the small Flag-F3-Myc protein, without success. This has included different gels, varying transfer times, different membranes and fixing on the membrane with glutaraldehyde or paraformaldehyde post-transfer.

To mitigate this, we have provided densitometry from three experiments performed independently at different times, with immunoblots from the additional two experiments provided in Supplementary Figure 8. In addition, we have included controls in which the F3 binding site on SOCS2 is mutated (L81F).

It is also worth noting that published data showing SOCS2 inhibition of GH responses (for example Greenhalgh et al., J Clin Invest 2005) is comparable to the data presented in Figure 5c-e, and in most cases is a modest inhibition, likely reflecting the mechanism of action of SOCS2, which is primarily proteasomal degradation of SH2-bound target proteins, or competition with other proteins for receptor binding. This is in contrast to SOCS1 and SOCS3 which are able to bind directly to JAK through a non-canonical SH2 interaction and inhibit JAK kinase activity through the SOCS1/3-KIR.

We have now included some additional comment in the results section (p13) as requested by the reviewer.

Overall the observation of the binding of a peptide to an exosite on SOCS2 SH2 is of interest and the characterisation of this is of general relevance as more evidence of non-canonical binding associated with SH2 domains is reported. Much of the analysis of the role of the F3 sequence in cells is over-interpreted and it is not really clear what the relevance of this is, particularly in the light of the fact that the sequence is not attributable to a known cellular ligand, or there is no proven therapeutic application of the use of F3.

We appreciate that SOCS2 inhibition of GH signalling is modest and have adjusted the language in the results section to better reflect this. We understand that without identification of an F3-like protein involved in regulating SOCS2 function, it is difficult to gauge the full relevance of this site. However, there are several ways to exploit this site and we have now expanded the discussion around how this could be of therapeutic benefit (p17). We have deliberately kept this discussion general, as we are indeed pursuing a commercial goal focused on this discovery and are not ready to outline our strategy in this publication.

Reviewer #3 (Remarks to the Author):

Linossi and colleagues have identified and targeted a novel allosteric binding site in the SH2 domain of SOCS2. The SOCS family proteins are key regulators of cytokine signaling with central roles in physiological signaling and in recent years critical functions were also identified in pathophysiological signaling in cancer and immunity. From a phage display library, the authors identified non-phosphorylated peptides that bound SOCS2 with low micromolar affinity. Excitingly, peptide F3 did not compete with phosphotyrosine (pY) ligand binding to the SOCS2 SH2 domain, but instead strongly enhanced pY peptide binding both in vitro and in pull-down assays from cells. Subsequent co-crystallization of F3 with a SOCS2-Elongin C-Elongin B complex showed an alpha helical conformation of F3 and a well-defined binding site, which was confirmed by extensive mutagenesis and NMR chemical shift perturbation mapping. The binding site is located between the N-terminal ESS helix (unique to the SOCS family) and the BC loop (adjacent to the pY binding pocket). SPR experiments then revealed the kinetic mechanism for enhanced binding to pY by F3 (mainly strongly reduced koff). Finally, the authors demonstrate enhanced SOCS2-dependent inhibition of growth hormone signaling in cells by expressing F3. This is an excellent piece of work that clearly demonstrates the presence of a previously unrecognized exosite that strongly regulates substrate recognition of the SOCS2 SH2 domain. This new exosite has not been previously identified despite more than a dozen crystal structures of SOCS family members. The study uses a wealth of quantitative biochemical and structural biology methods and confirmed results in cellular models. Experiments are well controlled and clearly presented and described. The findings are entirely novel and important evidence is presented that besides SOCS2, also the SOCS family member CISH could be regulated by ligands to this novel exosite. The findings are of great importance, as they indicate that substrate recognition by SOCS proteins could be strongly modulated by endogenous proteins that bind the novel exosite. Secondly, engineered ligands, such as F3, could be used therapeutically to enhance SOCS2-mediated inhibition of activated cytokine receptor signaling in various disease settings.

Minor points to be addressed:

1. Fig 1b: The ITC data for binding of F3, C4 and E11 should be shown in SI.

The ITC binding data for F3, C4 and E11 has now been included in Supplementary Fig. 9 and Supplementary Tables 1 & 2.

2. Table 1 reports binding affinities of mutants that were derived by either ITC or SPR. The Kd for F3 wt should be shown for both SPR and ITC to see if comparable values are obtained so that one can compare Kds of mutants obtained by the two different methods.

We agree and have now included the KD derived by SPR for WT F3 in Table 1. Table 1 and Table 2 also have now been updated to include data averaged from multiple ITC experiments and the mean-/+SD for SPR experiments.

3. line 107/108: ... truncations abrogate binding ... : Is the F3 peptide alone (not bound to SOCS2) alpha helical? And could the truncation result in loss or decreased degree of alpha helicality? This could be easily tested by far-UV CD spectroscopy.

We have performed NMR analysis of the labelled F3 peptide, which did not form an alpha helix in the absence of SOCS2. However, we did observe a propensity to form an alpha helix (not shown). Prediction algorithms also suggested F3 would

form a helical structure. Based on the co-crystal structure of the bound F3 it is clear that this truncation would disrupt the alpha helical structure. A comment has now been included in the discussion to address this point (page 15).

4. line 118/119: ... F3 ... did not compete with pY peptide from GHR: Where is this data shown? no reference to Fig. or SI Fig?

Our apologies for any lack of clarity. This experiment refers to the ITC data in Fig. 2a, where we observed enhancement of binding to phosphorylated Y595 peptide, rather than competition with phosphorylated Y595 peptide. An additional reference to Fig. 2a has now been included in the text.

5. line 171: typo in PDB entry: 6l4X and not 614X.

This has now been corrected.

6. line 224-256: It would be interesting to add thermal shift data to determine how much F3 binding may increase the melting temperature of SOCS2.

Thank-you for this suggestion. We have now performed thermal shift analysis showing that F3 binding increases the SOCS2 melting temperature by 5.8 degrees. This new data and methods have now been included in Supplementary Figure 2, and referenced in the Results section (p7).

7. Figure 5d and 5e lack anti-FLAG or anti-Myc immunoblot showing expression of the FLAG-F3-Myc construct. These should be added.

As commented on in our response to Reviewer 2, the detection of low molecular weight proteins by immunoblotting is a common issue in the field (i.e. Suzuki et al., Anal Biochem., 2008; <https://www.ptglab.com/news/blog/tech-tips-in-search-of-low-molecular-weight-proteins/>). We have spent considerable effort trying to detect the small Flag-F3-Myc protein, without success. This has included different gels, varying transfer times, different membranes and fixing on the membrane with glutaraldehyde or paraformaldehyde post-transfer. To mitigate experimental variation in transfection efficiency, we have provided densitometry from three experiments performed independently at different times, with immunoblots from the additional two experiments provided in Supplementary Figure 8.

8. The Discussion section rather vaguely speculated on SOCS2 as a possible therapeutic target. Some sentences should be added for which diseases and under which circumstances this might be useful. In addition, if and how F3 could be exploited therapeutically should be discussed: If F3 is alpha-helical when not bound to SOCS2, helix stapling and peptide delivery to cells could be attempted in a follow-up study.

We have now expanded the discussion around SOCS2 as a therapeutic target (page 17). As discussed above in response to point 3, the peptide is only alpha-helical when bound to SOCS2. However, given that we have observed some propensity for helix formation, there may still be merit in pursuing this approach in the future.

REVIEWERS' COMMENTS

Reviewer #2 (Remarks to the Author):

The authors have addressed all of the comments/suggestions in the original review. Although, the overall relevance of the work is limited by the lack of identification of the protein associated with F3 and the therapeutic potential is yet to be demonstrated, the work is suitable for publication in Nature Comms.